# Semi-Natural Superabsorbents Based on Starch-g-poly(acrylic acid): Modification, Synthesis and Application

**DOI:** 10.3390/polym12081794

**Published:** 2020-08-10

**Authors:** Elżbieta Czarnecka, Jacek Nowaczyk

**Affiliations:** 1Chair of Physical Chemistry and Physicochemistry of Polymers, Faculty of Chemistry, Nicolaus Copernicus University in Toruń, 7 Gagarina street, 87-100 Toruń, Poland; jacek.nowaczyk@umk.pl; 2Plastica Sp. z o.o., Frydrychowo 55, 87-410 Kowalewo Pomorskie, Poland

**Keywords:** starch, acrylic acid, superabsorbent, hydrogel, graft copolymerization

## Abstract

Biopolymer-based superabsorbent polymers (SAPs) are being synthesized and investigated as a biodegradable alternative for an entirely synthetic SAPs, particularly those based on acrylic acid and its derivatives. This article focuses on the chemical modification of starch (S), and synthesis of new potentially biodegradable polymers using acrylic acid (AA) as side chain monomer and crosslinking mediator together with *N*,*N*’-methylenebisacrylamide (MBA). The graft co-polymerization was initiated by ceric ammonium nitrate (CAN) or potassium persulfate (KPS), leading to different reaction mechanisms. For each of the initiators, three different synthetic routes were applied. The structures of new bio-based SAPs were characterized by means of IR spectroscopy. Thermogravimetric measurements were made to test the thermal stability, and morphology of the samples were examined using scanning electron microscopy (SEM). Physico-chemical measurements were performed to characterize properties of new materials such as swelling characteristics. The water absorption capacity of resulting hydrogels was measured in distilled water and 0.9% NaCl solution.

## 1. Introduction

The acrylic acid-based segment accounted for the largest share of 59.57% of the total volume in 2018. However, the polyacrylamide segment is expected to witness the fastest CAGR (Compound Annual Growth Rate) of 6.08% and the baby diaper segment will achieve CAGR (Compound Annual Growth Rate) of 6.01%, over the period of 2019 to 2024. The baby diapers segment accounted for the largest share of 70.38% of the total volume in 2018 [1].

The biocompatibility and high absorption capacity of hydrogels favor their use in tissue engineering [2], wound dressing [3], dental materials, cosmetics [4], drug delivery systems, and proteins [5]. In the agricultural sector, hydrogels are widely used, e.g., in the form of slow release systems for fertilizers [6], nutrients and water, or to fertilize the soil [7]. Additionally, hydrogels have been used in wastewater treatment [8] to remove toxic arsenate and Cr (VI) from aqueous solutions [9,10].

SAPs are three-dimensional polymeric networks capable of absorbing and holding an enormous amount of water or aqueous solutions. An important feature of these polymers is their ability to soak up water in relatively short time [3]. Superabsorbents have wide applications in a variety of fields, such as personal care, agricultural, forestry, chemical industry, and drug-delivery systems [11,12,13]. Superabsorbents manufactured using natural polymers have recently been extensively investigated due to their positive environmental impact. They are mainly based on polysaccharide hydrogels, and their most important advantage is biodegradability. Various SAPs based on natural polymers have been synthesized using cellulose [14,15], starch [16], chitosan [17,18,19], alginate [20], and gelatin [21]. Biodegradable polymers can be degraded by the enzymatic action of the microorganisms or their polymer chains can be broken down by processes such as chemical hydrolysis.

Starch is a natural polysaccharide, renewable, relatively cheap, abundant, and biodegradable polymer produced by plants as a reservoir of stored energy [22]. Starch is a blend of amylose and amylopectin, both consisting of α-d-glucose rings linked together via a α-d-(1,4) and/or α-d-(1,6) bonds. Amylose consists of long linear chains having sporadic side chains, in contrast to amylopectin molecules, which are heavily branched. Starch is found in staple crops such as potato, corn, rice, tapioca, wheat and in plant roots, crop seeds, and stalks [23]. It is a hydrophilic polymer containing a multitude of hydroxyl groups with a reactivity comparable to alcohols. Native starch shows a considerably low absorption ability and cannot be used directly as an absorbent. To improve absorption, it is necessary to modify native starch by introducing active groups with chelating and sorption abilities. Basically, there are four kinds of modifications mentioned in the literature: physical, chemical, genetical, and enzymatic [24,25]. 

Chemically modified starch shows a higher application potential and can be used more efficiently. The most common modifications of this polymer concern the esterification of hydroxyl groups or oxidative cleavage of C3-C4 bond of glucose ring. The modification is an important preliminary step of further graft copolymerization. The copolymerization following the C3–C4 bond cleavage in the starch polymers can be initiated with ceric salts and hydrogen peroxide. Further grafting reaction requires monomers such as acrylonitrile, acrylic acid, acrylate, and acrylamide. Usage of multifunctional (more than one C=C bond) acrylates will provide crosslinking necessary to the formation of 3D net structure important in hydrogels.

The most common method of synthesizing starch graft copolymers is the formation of active sites. Free radicals or ions are formed at a specific position in the polymer backbone. The network graft polymerization of starch and acrylic acid started with the decomposition of the initiator, resulting in the formation of free radicals [26]. The graft copolymerization of acrylic monomers onto natural polymers is almost exclusively performed by free radical polymerization. Among the chemical initiators, the ceric ion is the most selective initiator because it reacts directly with the starch backbone, forming free radicals at the point of attachment of the graft polymer. Due to its high cost, this initiator is not suitable for use on an industrial scale. In contrast, potassium persulfate decomposes on heating to form sulfate anionic radicals. The final step is a cross-linking agent to bind the acrylic acid chain and starch. Due to the presence of the cross-linking agent, a three-dimensional network was formed in the superabsorbent polymer system.

In gels, acrylic acid causes the formation of hydrogen bonds between the hydroxyl and carboxylic groups of starch. The reaction of acrylic acid with the initiator and gelatinized starch in water is a homogeneous system due to the excellent solubility of the discussed monomer. This will be advantageous in terms of responsiveness and ease of operation.

According to literature survey, most of the recent research in the field of biodegradable SAPs is focused on synthesis and modification of cellulose based polymers [27]. In the majority of solutions cellulose or modified cellulose chains are crosslinked and grafted using acrylic monomers [28]. Considerable interest in this approach turned our attention toward another abundant in nature polysaccharide, i.e., starch. In contrast to cellulose based SAPs starch based SAPs are seldom studied [29]. In the published studies concerning starch crosslinked with acrylic monomers, there is no information giving insight into aspects of polymer morphology influence on swelling properties. In our study, we have decided to emphasize on the part of material characterization concerning swelling properties of similar materials differing in morphology. In the course of this investigation, nine new SAP materials have been synthesized based on starch and acrylic acid. Application of nine different synthetic routes yielded morphologically different polymers. New materials were characterized structurally by means of infrared spectroscopy, structurally using electron microscopy technique and physio-chemically. The most important part of the characterization was the determination of swelling characteristics in water and artificial body fluids, which is of particular interest from the point of view of potential application as absorbent for hygiene products. The main goal of this investigation is the development of a methodology for synthesis of biodegradable SAPs based on starch as an alternative for non-degradable SAPs currently available. The study is part of a broader research of new biodegradable SAPs.

## 2. Methods of Characterization of Superabsorbents

### 2.1. Materials

Soluble starch (SS) ACS reagent grade (Sigma Aldrich, Poznań, Poland); corn starch (CS) pure (Sigma Aldrich, Poznań, Poland); acrylic acid (AA) pure (Sigma Aldrich); ceric ammonium nitrate (CAN) pure (Sigma Aldrich, Poznań, Poland); potassium persulfate analytical grade (Sigma Aldrich, Poznań, Poland); *N*,*N*’-methylenebisacrylamide (MBA) pure (Sigma Aldrich, Poznań, Poland); sodium hydroxide (NaOH) (Sigma Aldrich, Poznań, Poland); nitrogen gas (N_2_) technical grade; ethanol 96%_vol._ (Bioetanol AEG Ltd., Chełmża, Poland). The chemicals were used without further purification. All solutions were prepared using deionized water.

### 2.2. Fourier Transform Infrared Spectroscopy

Fourier transform infrared spectroscopy (FTIR) was used to identify the presence of specific chemical groups in the materials. FTIR spectra were obtained using a Bruker Vertex 70V spectrometer (Bruker Optoc GmbH, Ettlingen, Germany) in the wavenumber range from 4000 to 400 cm^−1^, for 16 scans with a resolution of 4 cm^−1^. FTIR spectra have been normalized and the main vibration bands have been associated with corresponding chemical groups. All spectra were analyzed using OPUS 7.5 software (Bruker Optoc GmbH, Ettlingen, Germany).

### 2.3. Thermal Analysis

Analyzes were carried out using a Simultaneous TGA-DTA thermal-analyzer type SDT 2960 Simultaneous TGA-DTA from TA Instruments, Champaign, IL, USA) at temperatures ranging from 20 °C to 1000 °C, at a heating rate of 10 °C·min^−1^, under atmospheric air with samples of ca. 2–4 mg. The total mass loss of the sample is equal to the peak area on the DTA curve. Recorded thermograms were analyzed using TA Universal Analysis Software.

### 2.4. Scanning Electron Microscopy

Surface topography and size of superabsorbent particles were tested using a scanning electron microscope manufactured by LEO Electron Microscopy Ltd. Cambridge, UK, model 1430 VP. Scanning electron microscopy was used to determine the shape, size, morphology, crosslinking density, and porosity of superabsorbents.

### 2.5. Preparation of Graft Polymerization with Ceric Ammonium Nitrate

The superabsorbents were obtained by graft polymerization of starch in aqueous solution using ceric ammonium nitrate as an initiating agent (Figure 1). Starch solution was prepared in a three-neck flask equipped with a magnetic stirrer, a condenser, and a thermometer. All processes were carried out under nitrogen atmosphere. The starch was prepared by stirring of 4.013 g starch powder in 100 mL deionized water at 95 °C. Afterwards, the starch solution was cooled down to room temperature. Then, 2 mL of CAN solution (0.005 mol/dm^3^) was poured into the solution in order to activate reactive sites in the starch backbone and initiate free radical polymerization of acrylic monomers [30,31]. AA (5.998 g) monomer neutralized beforehand with NaOH was added as grafting monomer and after that 0.05% (relative to monomer) of MBA was added as crosslink agent. The mixture was stirred and kept for 2 h at 70 °C. Afterwards, the system was cooled down to room temperature. The product was washed with ethanol and distilled water and dried in a vacuum oven (40 °C) to constant weight. This product has been given an abbreviated name SS-g-PAA(CAN).

Another superabsorbent was made in the same way, except that this time the gelatinized starch obtained by seasoning the solution for 12 h at 4 °C and later used in the synthesis yielded with a product called SS(12)-g-PAA(CAN).

The third superabsorbent was made in the same way, with the difference being that the precipitated end product was left in the dark for 12 h in a solution of deionized water and ethanol and later separated from post reaction mixture. The product was called SS-g-PAA(12/CAN).

### 2.6. Preparation of Starch Graft Copolymer Using Potassium Persulfate Initiator

Another set of superabsorbents was obtained by graft polymerization of starch using KPS as an initiating agent (Figure 2). The starch solution was prepared according to the same procedure as in previously described examples. After cooling to room temperature, KPS was added followed by addition of AA neutralized with NaOH and then MBA [32]. The system was cooled to room temperature and pH was adjusted to 10 with acetic acid. The mixture was stirred for 3 h under N_2_ atmosphere and at 70 °C. The product was washed with ethanol and deionized water and then dried. This product was named as SS-g-PAA (KPS).

### 2.7. Preparation Synthesis with Urea/NaOH

The graft copolymerization was performed in beaker equipped with stirrer. Corn starch (1.509 g) was dispersed in 27 mL of deionized water, and then 18 mL of 4.509 g NaOH solution was added dropwise. After the starch was fully gelatinized, 12 mL of 3.007 g urea solution and 0.153 g KPS or 0.300 g CAN in aqueous solution was added subsequently and stirred for 30 min at room temperature. Then, 0.063 g MBA and 4.498 g AA dissolved in 4.5 mL of deionized water were added in sequence into the stirred mixture and finally capped and maintained at 65 °C for 5 h. The reaction mixture was cooled to room temperature, and a hydrogel was precipitated and washed with ethanol-water mixture (7:3 wt) [33]. Subsequently, it was dehydrated by 95% and anhydrous ethanol and then dried at 40 °C for 48 h under vacuum oven. The regenerated starch was ground [34]. These products were named as CS-g-PAA (KPS/U/NaOH) or CS-g-PAA (CAN/U/NaOH), respectively.

The polymer sample codes related to different synthesis methods are summarized in Table 1. 

### 2.8. Swelling Properties

To determine the gel content values, 0.100 g of dried sample was dispersed in double distilled water to swell for 72 h. After filtration, the extracted gel was dewatered by ethanol, dried for 24 h at 40 °C, and then reweighed. Gel content (gel %) was calculated using following formula Equation (1):(1)$$Gel\;\left[\%\right]=\frac{m}{m_{0}}\times 100,$$
where, *m* and *m*_0_ stand for final and initial weight of sample, respectively.

The equilibrium water absorption of the superabsorbent was calculated using following equation (Equation (2)):(2)$$Q_{eq}\;\left[\frac{g}{g}\right]=\frac{w_{s}-w_{d}}{w_{d}},$$
where *Q_eq_* [g/g] is the equilibrium water absorption calculated as grams of water per gram of sample, *w_d_* and *w_s_* are the weights of the dry sample and water swollen sample, respectively.

### 2.9. Swelling Kinetics and Effect of Particle Size on Swelling

Water absorption and water retention of superabsorbent hydrogels with different particle sizes were determined according to methods widely described in the literature. The study of absorbency rate of the synthesized hydrogels, approximately (0.100 ± 0.001 g) with various particle sizes were poured into weighted tea bags (270 mesh) and immersed in 250 mL distilled water. The water absorbency (*Q_t_*) was measured as a function of time (*t*). The swelling behavior of SAP samples were studied by using salt solutions (0.9 wt% NaCl solution).

## 3. Results and Discussion

### 3.1. Analysis of the Synthesis Mechanism

Two main reactions for the production of polysaccharide-based hydrogels can be distinguished: graft copolymerization of vinyl monomers on a polysaccharide in the presence of a cross-linking agent and a direct cross-linking reaction of the polysaccharide. The first of these methods is used in this publication.

OH groups derived from saccharide units interact with Ce^4+^ ions to form complexes based on redox pairs. The next step is the cleavage of C_2_-C_3_ bonds of the saccharide units as a result of the action of carbon radicals formed as a result of the dissociation of the complexes. The resulting free radicals initiate the graft polymerization of the crosslinker and vinyl monomers on the starch chains. The formation of hydrogels by graft copolymerization of acrylic acid on corn starch was also analyzed in Athawale et al. [30]. The publication by Dragan and Apopea presents the results of the research and the mechanism of grafting acrylamide onto starch with Ce^4+^ as a free radical initiator [35]. The same mechanism was proposed for the grafting reaction of acrylic acid onto starch in Figure 1.

As a result of the action of potassium persulfate as an initiator on polysaccharide chains, free hydrogen radicals from OH groups are detached. In this method, it is important to maintain a certain temperature (much higher than in the case of Ce^4+^) because a thermal initiator was used. As a result of the action of potassium persulfate as an initiator on polysaccharide chains, free hydrogen radicals from OH groups are detached. In this method, it is important to maintain a certain temperature (twice as high as for Ce^4+^) because a thermal initiator is used. In the publication of Pourjavadi et al. ammonium persulfate was used as an initiator of the graft copolymerization reaction of acrylic acid with kappa-carrageenan in the presence of a cross-linking agent in the form of *N*,*N*’-methylenebisacrylamide [36]. The persulfate initiator decomposes on heating to form the sulfate anion. The radical cleaves hydrogen from the starch hydroxyl groups, forming alkoxy radicals on the starch chains.

In the case of direct cross-linking of polysaccharides, polyfunctional or polyvinyl compounds are used, however, this method will be described in subsequent papers, as research is currently underway.

### 3.2. FTIR Analysis

FTIR spectra of all monomers grafted on starch (corn starch, soluble starch) were quite similar to each other. In the spectrum of starch, the transmittance bands at 3312 cm^−1^, 573 cm^−1^, 528 cm^−1^ are attributed to the OH group and a smaller band at 2928 cm^−1^ to C–H stretching vibration [37]. The pattern of the signals changes after polymerization, which indicated that graft copolymerization occurred between the AA monomer and the molecules SS and CS. The PAA-grafted chains showed a new band at about 1700 cm^−1^ characteristic for C=O stretching. The most significant notion derived from juxtaposition of the FTIR spectra of four grafted starch gels in Figure 3 is the presence of several peaks at the same position, which indicates that the resulting materials contain moieties derived from starch, acrylic acid, and MBA included in its structure. Moreover, FTIR spectra of the gels show no signals indicating the presence of C=C groups, which proves that the AA in free form is not present in the material, and all the monomer undergone graft copolymerization. All spectra with designated characteristic peaks were situated in the Appendix A.

Analyzing the spectra of superabsorbent obtained in presence of urea, no significant differences in spectra were observed in comparison with that of starch-AA without urea. Peaks appeared at 3303 cm^−1^, 2922 cm^−1^, 1704 cm^−1^, 1558 cm^−1^, 1447 cm^−1^, and 1405 cm^−1^. Additionally, in comparison with the IR spectrum of native starch, similar absorption peaks appeared in 1151 cm^−1^, 1078 cm^−1^, 993 cm^−1^, and 930 cm^−1^, which are the characteristic peaks of starch structure. The C–O–C symmetrical stretching, C-OH bending and the skeletal vibration of the pyranose ring are characterized by peaks at 928 cm^−1^, 851 cm^−1^, and 573 cm^−1^ [33]. The intensities of all characteristic bands decreased after the polymerization process, which proves that the copolymerization between AA and starch was successful [38]. A remarkable change in FTIR spectra of starch-based copolymers was the appearance of the peak at 1700–1728 cm^−1^, which is regarded to C=O vibrations in aldehyde groups [39].

The largest changes were observed for material prepared instantly after initial solution preparation (SS-g-AA (CAN)), while the smallest differences were noted for the product obtained from the starch solution seasoned by 12 h (SS-g-AA (12) (CAN)).

### 3.3. Scanning Electron Microscope 

The pore size of the superabsorbents mainly depends on the many factors affecting the swelling and micromorphology of hydrogels. The large differences in the surface morphology of starch copolymers are shown in Figure 4 (original size SEM images are included in Appendix A). The starch copolymers prepared in urea/sodium hydroxide solution: CS-g-PAA (CAN/U/NaOH) (c) and CS-g-PAA (KPS/U/NaOH) (d) presents the largest variation in morphology and pore size compared with the other compositions. These polymers form agglomerates resembling bushes. Their structure is looser and highly porous with irregular voids appearing between agglomerates. The average diameter of the holes or interstices is ranging from 11 to 136 μm and the average is 57 μm.

The hydrogels containing low crosslinking density generally form structures with wider pores. Those containing high crosslinking density are randomly aggregated and exhibit more granular-like structure. This is due to the differences in the expansion of polymer network and in the magnitude of the affinity of the polymer network for water. Additionally, the decrease in pore size can be attributed to the increase in the elasticity of the hydrogels related to crosslinking density.

In the literature, it was found that CS and SS hydrogels occur in the form of variable size irregular granules having smooth surface [30,40]. After binding acrylic acid into the starch backbone, the surface roughness increased, and some folds can be observed (Figure 4c–g). The material has a higher specific surface area when the surface of the hydrogel composite is folded. These folds may be regions where water or other external stimuli can interact with the hydrophilic groups of the grafted copolymers and permeate, which facilitated the permeation of water into the polymeric network [41]. Regarding micrographs e) SS-g-PAA (CAN), (f) SS(12)-g-PAA (CAN), g) SS-g-PAA (12/CAN), and h) SS-g-PAA (KPS), all materials were found to be granular, with divergent particles size and shape. The majority of the granules showed oval shape and rather smooth surface. Observed occasionally truncations and slight erosion marks can be attributed to either mechanical damage of the granules during the extraction process or partial oxidation of the surface.

The granule size distributions for both starch kinds revealed a bimodal feature with a lot of small and large particles, but not many medium-sized. The small-size granules had diameters ranging between 6 to 10 µm. Starch (CS and SS) particles have oval shapes with smooth surfaces, unlike acrylic acid modified products. The modification not only changes the surface, but also leads to the formation of particles of larger size with diameter between the 250 and 600 µm.

In the case of CS-g-PAA (CAN/U/NaOH) (c) and CS-g-PAA (KPS/U/NaOH) (d), more compact and porous structures were found in the micrographs that can be attributed to the granules’ agglomeration. The presence of spacious pores guarantees a reduction in transport resistance and an increase in the total water sorption capacity [41,42]. The channels, cracks, and pores in the copolymers create accessible spaces for the extensive hydration of hydrophilic groups of the material responsible for swelling.

### 3.4. Thermogravimetric Analysis

TGA methods were used to investigate thermal behavior of starch and copolymers obtained by grafting acrylic monomers. All the recorded thermograms are included in Appendix A and only the most important data are shown in the Table 2 and Table 3. Starch shows a typical two–steps thermogram. The first step shows weight loss at about 5.0%, in 70.62 °C for CS and 59.24 °C for SS. In the second step, the main weight loss occurs for CS (54.88%) at 313.85 °C, and for SS (77.82%) at 229.49 °C [43]. Based on this data, we can assume that corn starch is more thermally stable than soluble starch. The thermogram of AA show a multi-step degradation process with the lack of weight loss up to 100 °C except for a modest weight loss between 80 and 100 °C found for each sample investigated in this study. There are three to seven decomposition stages on the thermograms of (SS-g-PAA(12/CAN), CS-g-PAA(CAN/U/NaOH)), and (SS-g-PAA(CAN)). The loss of water captured in the polymer network can be attributed to the weight loss below 200 °C. The maximum weight loss was recorded in the temperature range between 250 and 500 °C, which may be explained by the degradation of starch backbone of the superabsorbent. The weight loss found above 500 °C, can be ascribed to degradation of the polymer chains and acrylate crosslinked grafts, which shows that the thermal stability of the starch copolymers is higher than that of native starch. Additionally, the addition of AA and other synthetic monomers to starch improves its thermal stability.

Analyzing SS-g-PAA (CAN) and SS-g-PAA (KPS) one can notice an additional weight loss stage (~10%) at 260–270 °C, which is caused by the breakdown of starch. The next stage of decomposition in the temperature range between ~260–340 °C is ~35–45% weight loss associated with the degradation of the starch backbone in each sample. Starch pyrolysis is the main cause of weight loss, which is associated to the formation of ether segments and dehydration of its chains, elimination of CO and CO_2_ and random chain scission [44]. The greater the ratio of amylopectin to amylose improves the thermal stability of starch-based superabsorbents, because the resulting additional glycosidic bonds in the amylopectin molecule increase the thermal stability of the copolymers compared to the linear structure of amylose [45]. For CS-g-PAA (CAN / U / NaOH) and CS-g-PAA (KPS / U / NaOH) the highest weight loss (45%) was observed at the decomposition temperature of 292.54 and 340.49 °C. These peaks are attributed to oxidation and thermal decomposition, but peaks at 452.38 and 432.84 °C may be due to carbonate formation. The product with the addition of urea has better thermal stability due to nitrogen.

### 3.5. Swelling Properties of Superabsorbents

In personal hygiene products, the important feature is the swelling ability in salt solutions. The grafting of acrylic acid onto starch produces polymer chains with a large number of negatively charged groups, e.g., –COO¯. Although carboxyl groups are accompanied by cationic counter-ions, the negative charge groups repel each other yielding in chains stretching. This provides extra space in the polymer network that can occupied by water or aqueous solution entering the polymer network to level the osmotic pressure. Consequently, polymer absorbs and retains a large volume of water or aqueous solutions. The insolubility of superabsorbent in aqueous solutions is achieved by cross-linking with *N*,*N*-methylenebisacrylamide.

The screening effect of positive cation charges is the reason for the decrease in the degree of swelling of superabsorbents in salt solutions compared to the absorption of pure water (Figure 5.). Electrostatic repulsion between anions present in the system leads to a decrease in the osmotic pressure between the hydrogel internal solution and the external solution. Salt solutions with polyvalent cations reduce the swelling of hydrogel more than univalent cations, moreover, multivalent cations can cause so called ionic crosslinking of the polymer matrix. In consequence, the gel absorption significantly decreases with increase of cations valence [46,47].

The increase in the ionic strength of the salt solution reduces the absorption of deionized water. Therefore, the experimental results abide by Flory’s equation [48,49] (Equation (3)):(3)$$Swelling^{\frac{5}{3}}≅\left(v\;M_{c}\right)-1\;{\left(\frac{Mc}{M}\right)}^{-1}\frac{\left(\frac{1}{2}-x_{1}\right)}{v_{1}},$$
where

*v*—the specific volume of the polymer [m^3^·kg^−1^],

*M_c_*—the chain length between crosslinks [n],

*M*—primary molecular mass [u],

*x*_1_—the Flory solvent-polymer interaction term [-],

*v*_1_—the volume fraction of the polymer [-].

The addition of urea to the mixture during synthesis has a significant impact on the water absorption capacity of polymers, as is the case with NaOH. At the presence of water in elevated temperature urea molecules can undergo hydrolysis, liberating NH_4_^+^ and CNO^−^ ions [50]. The main purpose of adding urea was to increase the extent of the starch molecule chains in the solution.

The high swelling ability is of great importance in the practical application of a hydrogel, while the swelling speed is particularly important in superabsorbent applications. The swelling kinetics of superabsorbents are significantly influenced by factors such as swelling capacity, powder particle size distribution, specific surface area, and polymer composition [51,52]. The plots in Figure 6 show the dynamics of swelling of a superabsorbent sample with certain particle sizes (larger than 40 meshes, 40–60 meshes and smaller than 60 meshes) in water. Initially, a sharp increase in swelling rate was noted, while the longer, the more flattened the line until the process equilibrium value was reached. The initial swelling rate can be calculated using an equation based on Voigt [51] (Equation (1)),
(4)$$S_{t}=S_{e}\;\left(1-e^{-\frac{t}{T}}\right),$$
where: *S_t_* ($\frac{g}{g}$) - swelling at time *t (min)*, *S_e_* ($\frac{g}{g}$) *-* equilibrium swelling, t (min) – time, τ (min) - “rate parameter”.

Table 4 presents the rate parameters for superabsorbents with different particle sizes. Since τ is a measure of the resistance to water permeation, the lower the τ value, the higher the rate of water uptake will be. The lower the particles size, the higher the rate of water uptake, which is closely related to superabsorbent surface area to volume ratio. Since the uptake rate depends on the diffusion process it is obvious that the increase of the contact area will improve the diffusion rate [53]. Another important aspect is of structural nature and involves the degree of crosslinking, which influences both the swelling capacity and sorption rate. The crosslinking is important to prevent the polymer dissolution, however a high crosslinking degree makes the polymer more compact and less susceptible to swelling.

The complex kinetics of SAP swelling is related to a specific interaction between the swelling medium molecules and the polymer side groups and the structural topology of the polymeric 3D network responsible for susceptibility to formation of voids that can be filled with the medium [54].

According to the results presented in this article, hydrogels based on acrylic acid and starch show good absorption properties, high thermal capacity, and reusability. These properties require further improvement, which should be considered in future research works. The presented work includes preliminary studies for a large research project with the main goal of developing a biodegradable superabsorbent for use in the hygiene sector. Low cost, biodegradable properties, and non-toxicity make starch hydrogels attractive subjects of investigation due to their ability to absorb large amounts of water as well as heavy metals and dyes. Hydrogels are widely used in everyday products, although their potential has not yet been fully explored. These materials are known in the market for wound dressings, contact lenses, and hygiene products. Commercial hydrogel products in tissue engineering and drug delivery are still limited, despite numerous publications, projects and patents that have been produced. The high cost of production is a major limitation in the production of the discussed hydrogels on an industrial scale.

There are preliminary studies, so for us the most important conclusion is that the synthesis process should be modified and improved, and a better superabsorbent composition should be developed so that the degree of liquid absorption is much higher.

## 4. Conclusions

Significant research effort is underway around the world to obtain new materials, that will combine raw material obtained from natural, renewable sources and, biodegradability. This is extremely important for new superabsorbent materials for personal hygiene products such as disposable diapers, which are responsible for serious environmental issues. Several starch-based superabsorbent co-polymers using acrylic acid were prepared by copolymerization via free radical grafting. The syntheses were carried out in different medium including deionized water, sodium hydroxide solutions, and urea solutions. Potassium persulphate and ceric ammonium nitrate (IV) were used as initiators, while hydrophilic *N*,*N*’-methylenebisacrylamide was used as the crosslinking agent. Comparing the effect of discussed initiators, it can be concluded that the persulfate oxidation capacity is weaker, the reaction rate is lower, the reaction time is longer, and the reaction temperature is higher. FTIR analysis and thermogravimetric analysis revealed that AA monomers were grafted to macromolecular chains of CS or SS. The swelling capacity of hydrogels depends on the concentration of crosslinker (MBA) and the monomer ratio, such that swelling is reduced by increasing the MBA concentration. The decrease in absorption capacity in salt solutions compared to distilled water can be attributed to the effect of screening charge and ion crosslinking for mono and multivalent cations respectively. The obtained results confirmed the assumptions that the smaller the particle size, the faster the liquid absorption rate. Materials synthesized with sodium hydroxide and urea solution showed the best absorption of salty water.

The synthesized materials can be considered as an effective alternative to improve the moisture content of various soils for agricultural use. Moreover, in the future, such materials can be used as ion exchangers to increase their efficiency in wastewater treatment. Efforts to improve the property profile of cross-linked hydrogels are still ongoing, and we hope that the overall performance in terms of water absorption capacity and aqueous solutions will further improve.

## Figures and Tables

**Figure 1 polymers-12-01794-f001:**
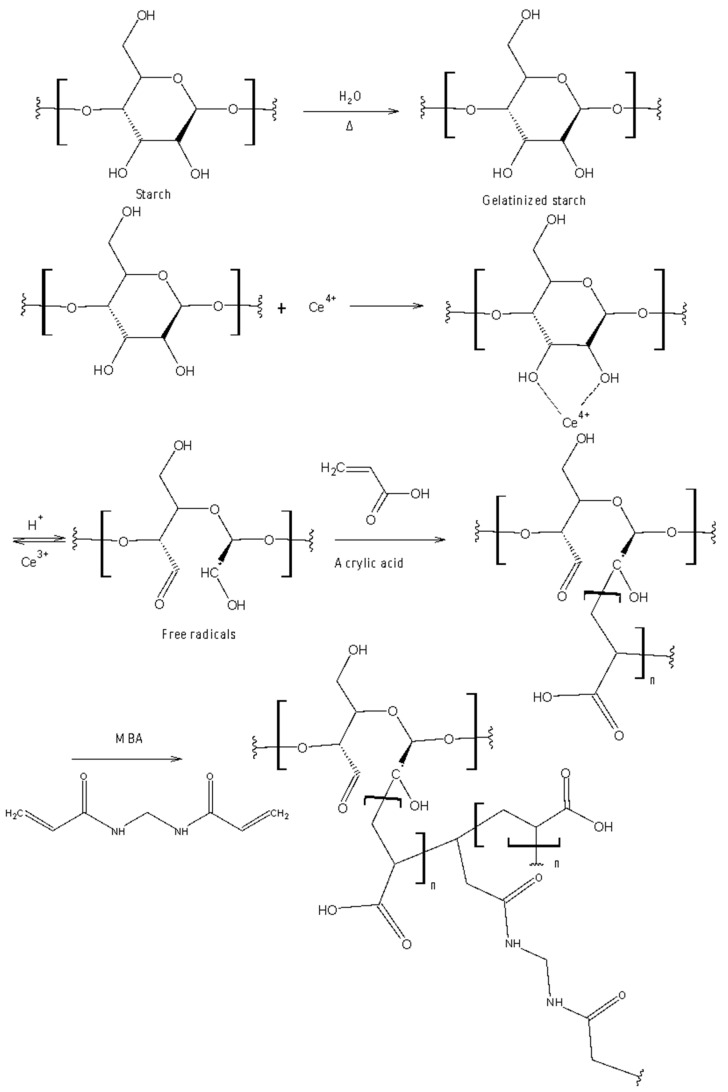
Mechanism of the crosslinking reaction of starch and acrylic acid initiated by ceric IV ions as a crosslinking agent.

**Figure 2 polymers-12-01794-f002:**
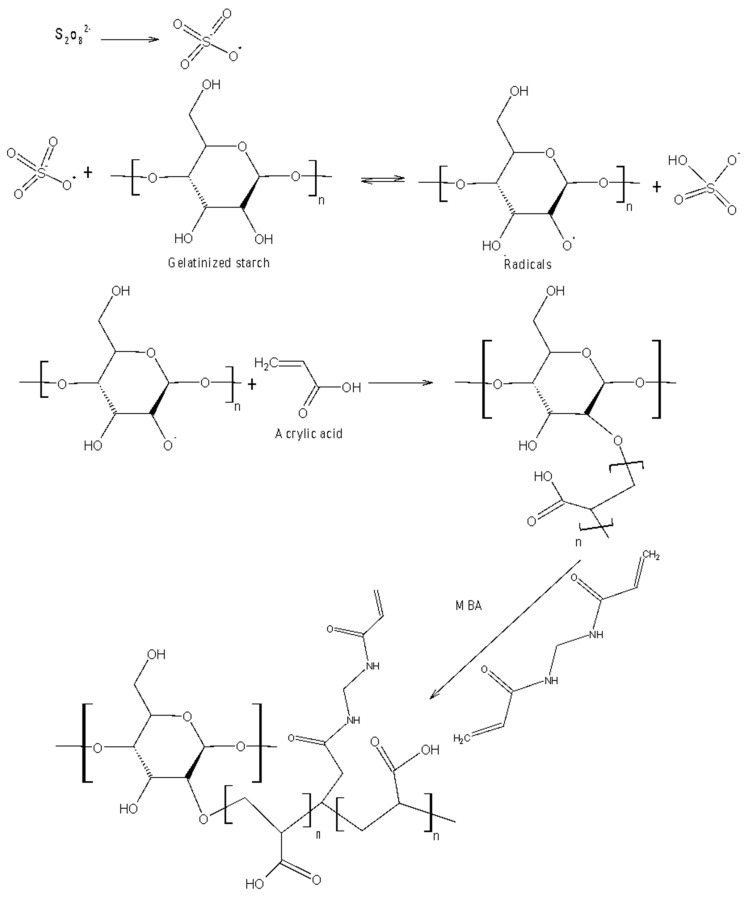
Mechanism of the crosslinking reaction of starch and acrylic acid with potassium persulfate as a crosslinking agent.

**Figure 3 polymers-12-01794-f003:**
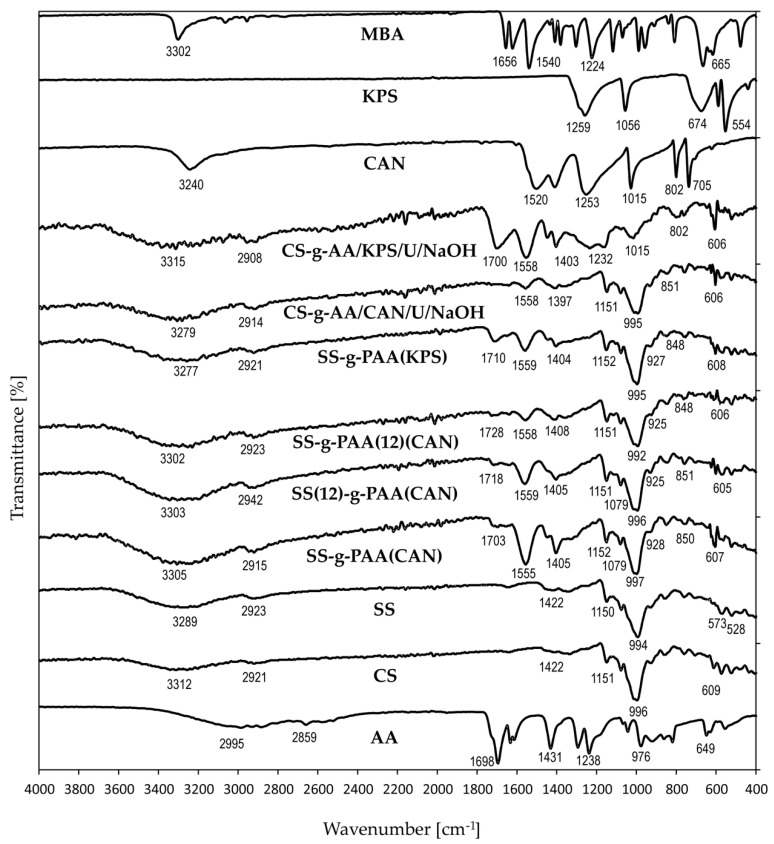
FTIR spectra of SS, SS-g-PAA (12)(CAN), SS(12)-g-PAA (CAN), SS-g-PAA (CAN), SS-g-PAA (KPS), CS, CS-g-PAA (KPS), CS-g-PAA (U/NaOH/CAN), AA and MBA.

**Figure 4 polymers-12-01794-f004:**
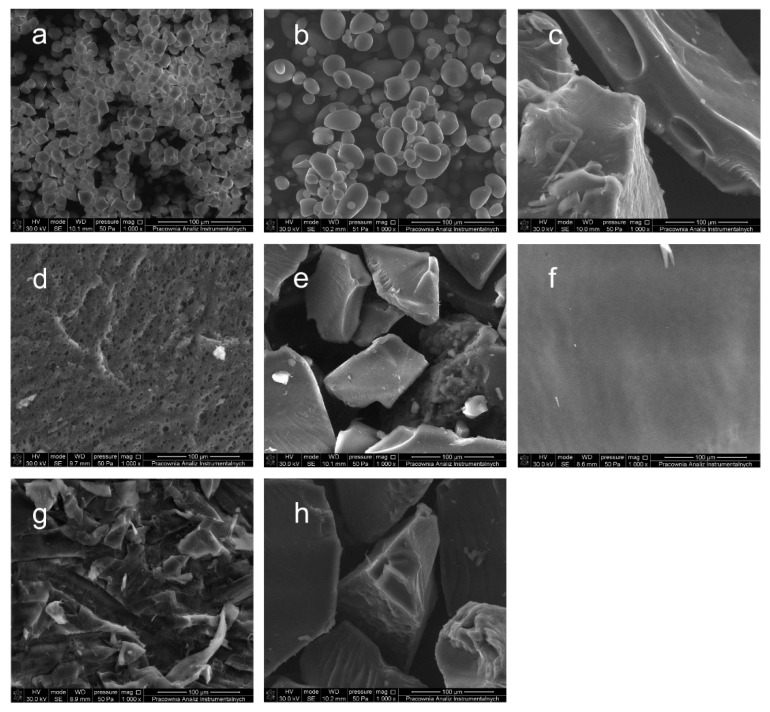
SEM images at 1000× magnification of substances: (**a**) corn starch, (**b**) solvent starch, (**c**) CS-g-PAA (CAN/U/NaOH), (**d**) CS-g-PAA (KPS/U/NaOH), (**e**) SS-g-PAA (CAN), (**f**) SS(12)-g-PAA (CAN), (**g**) SS-g-PAA (12/CAN), (**h**) SS-g-PAA (KPS).

**Figure 5 polymers-12-01794-f005:**
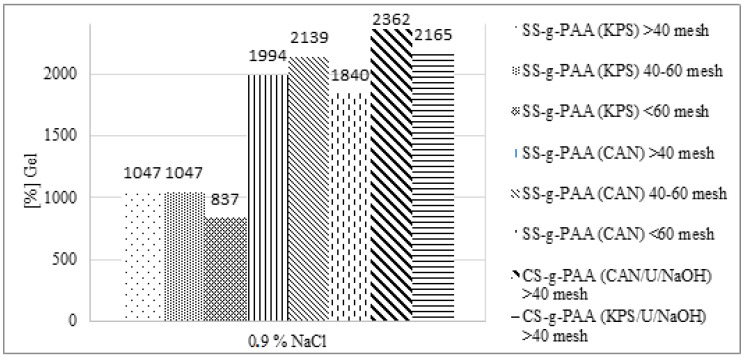
Swelling capacity of superabsorbents in 0.9 wt% NaCl_aq_ solution (**A**) percentage of gel in all samples, (**B**) saline absorbency of samples prepared with potassium persulfate in a given time for different grain sizes, (**C**) saline absorbency of samples prepared with ceric ammonium nitrate in a given time for different grain sizes.

**Figure 6 polymers-12-01794-f006:**
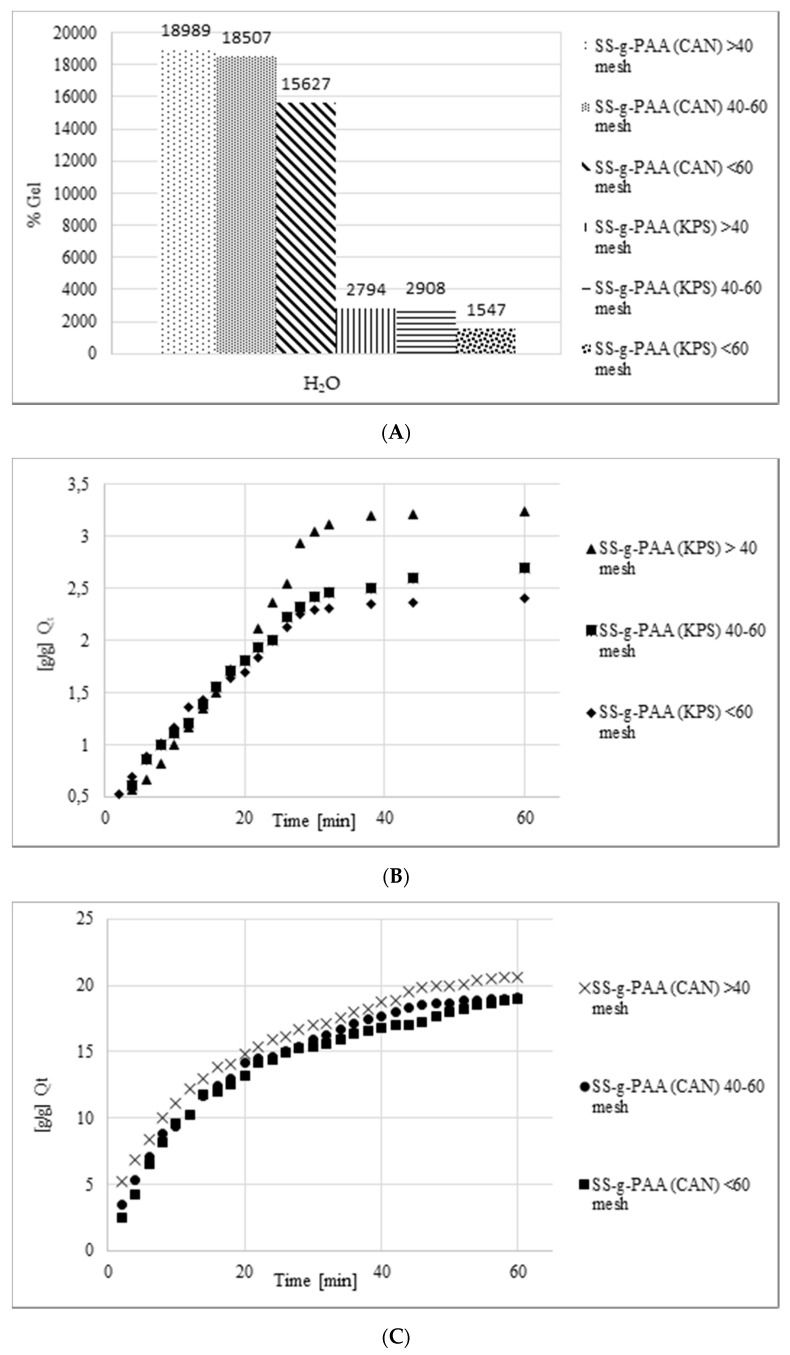
Swelling capacity of superabsorbents in dejonized water: (**A**) percentage of gel in all samples, (**B**) water absorbency of samples prepared with potassium persulfate in a given time for different grain sizes, (**C**) water absorbency of samples prepared with ceric ammonium nitrate in a given time for different grain sizes.

**Table 1 polymers-12-01794-t001:** Explanation of polymer sample codes.

Samples Code	Description
SS	Soluble starch
CS	Corn starch
CAN	Ceric ammonium nitrate
KPS	Potassium persulfate
MBA	*N*,*N*’-methylenebisacrylamide
SS-g-PAA(12/CAN)	Soluble starch-g-poly(acrylic acid) with ceric ammonium nitrate and the precipitated end product was left in the dark for 12 h in a solution of deionized water and ethanol and later separated from post reaction mixture.
SS(12)-g-PAA (CAN)	Soluble starch-g-poly(acrylic acid) with ceric ammonium nitrate and the superabsorbent was cooled for 12 h at 4 °C and then precipitated.
SS-g-PAA (CAN)	Soluble starch-g-poly(acrylic acid) with ceric ammonium nitrate and the product was washed with ethanol and distilled water immediately.
SS-g-PAA (KPS)	Soluble starch-g-poly(acrylic acid) with potassium persulfate and the product was washed with ethanol and distilled water immediately.
CS-g-PAA (CAN/U/NaOH)	Corn starch-g-poly(acrylic acid) with ceric ammonium nitrate and the product was prepared in deionized water and NaOH solution.
CS-g-PAA (KPS/U/NaOH)	Corn starch-g-poly(acrylic acid) with potassium persulfate and the product was prepared in deionized water and NaOH solution.

**Table 2 polymers-12-01794-t002:** The mass loss results derived from thermogravimetric analysis (TGA).

Samples	TGA (5 wt % Loss) [°C]	TGA (10 wt % Loss) [°C]	TGA (50 wt % Loss) [°C]
SS	59.24	105.44	355.92
CS	70.62	280.23	315.80
CAN	209.77	211.97	220.87
KPS	295.48	472.73	
MBA	198.50	210.74	250.24
SS-g-PAA(12/CAN)	139.04	244.64	328.84
SS(12)-g-PAA	142.25	232.38	362.80
SS-g-PAA (CAN)	97.26	174.90	451.69
CS-g-PAA (CAN/U/NaOH)	118.26	254.35	315.37
SS-g-PAA (KPS)	112.73	214.19	384.57
CS-g-PAA(KPS/U/NaOH)	137.02	193.60	340.02

**Table 3 polymers-12-01794-t003:** The specific decomposition temperatures read from DTA plots of the polymers.

Sample Code	Decomposition Temperature [°C]						
	Step 1	Step 2	Step 3	Step 4	Step 5	Step 6	Step 7
SS	59.00	229.49	361.80	-	-	-	-
CS	60.77	313.85	-	-	-	-	-
CAN	200.03	221.88	257.04	-	-	-	-
KPS	237.48	294.32	321.84	532.30	590.91	805.81	-
MBA	187.76	239.26	333.39	348.48	364.47	-	-
SS-g-PAA(12/CAN)	121.16	290.76	451.49	-	-	-	-
SS(12)-g-PAA (CAN)	128.26	259.68	288.99	447.94	912.37	-	-
SS-g-PAA (CAN)	66.99	273.00	291.65	344.93	452.38	732.10	814.69
CS-g-PAA	88.30	292.54	452.38	-	-	-	-
SS-g-PAA (KPS)	67.88	267.68	293.43	356.43	448.83	787.16	-
CS-g-PAA(KPS/U/NaOH)	193.08	296.09	340.49	432.84	804.92	-	-

**Table 4 polymers-12-01794-t004:** The rate parameters for superabsorbent with different particle sizes.

τ [min] for SS-g-PAA (KPS)			τ [min] for SS-g-PAA (CAN)		
<60 mesh	40–60 mesh	>40 mesh	<60 mesh	40−60 mesh	>40 mesh
17.04	23.42	30.03	11.83	15.80	18.73

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
