# Peer review of "Semi-Natural Superabsorbents Based on Starch-g-poly(acrylic acid): Modification, Synthesis and Application"

_polymers, 2020, doi:10.3390/polym12081794_

Round 1

Reviewer 1 Report

In this manuscript, the authors reported the modification, synthesis, and application of superabsorbent polymers (SAP) based on acrylic acid and its derivatives. The synthesized bio-based SAPs were characterized with multi-techniques and utilized for the evaluation of the absorption capacity of the resulted hydrogels in distilled water and NaCl solution. It is an interesting work, and this work will be helpful for the synthesis of novel polymer nanomaterials for environmental and biomedical application. This manuscript could be accepted for publication at Polymers after major revisions.

Special comments for the revisions:

  1. In the “Introduction” part, it is suggested for the authors to add more information on the synthesis and application of polymer hydrogels. In addition, the authors should make it more clear for the novelty and significance of this work.
  2. It is suggested for the authors to add a scheme to indicate the synthesis process of this novel SAP materials.
  3. In Part 2, it is suggested for the authors to add a new section to present the information of the characterization techniques used in this work.
  4. In Figure 1, the authors are suggested to label some key peaks in the figure.
  5. In Figure 2, the scale bars are hard to see.
  6. What are the potential applications of the synthesized SAPs? It is necessary for the authors to add more discussion to make it more clear.
  7. How to prove the high absorption performance of the synthesized polymer hydrogels? more discussion is needed.

Author Response

In this manuscript, the authors reported the modification, synthesis, and application of superabsorbent polymers (SAP) based on acrylic acid and its derivatives. The synthesized bio-based SAPs were characterized with multi-techniques and utilized for the evaluation of the absorption capacity of the resulted hydrogels in distilled water and NaCl solution. It is an interesting work, and this work will be helpful for the synthesis of novel polymer nanomaterials for environmental and biomedical application. This manuscript could be accepted for publication at Polymers after major revisions.

  1. In the “Introduction” part, it is suggested for the authors to add more information on the synthesis and application of polymer hydrogels. In addition, the authors should make it more clear for the novelty and significance of this work.

Answer: The whole manuscript has been read carefully again and some parts have been deleted. The introduction was extended with information on the synthesis and application of polymer hydrogels, and the novelty and significance of this work was explained, as suggested by the reviewer. All changes suggested by the Reviewer are marked green, while additional changes are marked in blue.

  1. It is suggested for the authors to add a scheme to indicate the synthesis process of this novel SAP materials.

Answer: Schemes of reaction mechanisms have been added in sections 2.5 and 2.6 of the section describing the methodology of synthesis using specific initiators.

  1. In Part 2, it is suggested for the authors to add a new section to present the information of the characterization techniques used in this work.

Answer: This chapter has been developed as suggested by the Reviewer. Three new subsections (2.2, 2.3, 2.4) describing techniques for superabsorbent characterization have been added.

  1. In Figure 1, the authors are suggested to label some key peaks in the figure.

Answer: In Figure 3 (new numbering after introduced changes) several wavelengths for the most characteristic bands have been marked. All the spectra of the discussed samples were additionally included in the Supplementary Materials (Figures S1 to S22) along with the determined wavelengths characteristic for a given sample.

  1. In Figure 2, the scale bars are hard to see.

Answer: Figure 2 has been modified by deleting some of the photos, replacing them with new, larger ones with a higher resolution and x100 magnification. In addition, several other photos of different resolution were placed in the Supplementary Materials.

  1. What are the potential applications of the synthesized SAPs? It is necessary for the authors to add more discussion to make it more clear.

Answer: A new paragraph has been added to the end of the Results and Discussion section, the potential use of the materials obtained.

  1. How to prove the high absorption performance of the synthesized polymer hydrogels? more discussion is needed.

Answer: Absorption performance is rather a comparative quantity and It has no defined standard, since that we can only measure swelling capacity and rate and compared them with other measured systems in the same conditions.  And this is what we have performed in or study. The results of comparison between all 9 polymer samples is provided in the manuscript. If more discussion on this topic is necessary, we kindly as the referee to specify what issues we wave missed in our current discussion.

Reviewer 2 Report

Dear Authors,

The manuscript needs major revision.

Extensive editing of English language and style required. Many sentences must be reformulated.

Many experimental details must be added (characterization methods; thermograms).

The legend of figures must be revised.

FTIR  and SEM figures must be revised as suggested.

The authors confused urea with urine.

Authors must give the correct meanings of all physical quantities in equation 3.

Authors must be consequent with the abbreviations (FTIR or FT-IR, etc.).

Authors mentioned that they obtained nanocomposites, but they commented SEM images of hydrogels obtained and concluded that they have dimensions
ranging between 6 and 600 micrometers, so it is a contradiction. Authors must revise this aspect.

Authors must pay attention to all yellow highlighted paragraphs/ words.

All the sections must be revised.

I made some corrections/suggestions/comments in the attached manuscript.

Author Response

On behalf of my co-authors and myself I want to express our thanks to the Reviewers for the valuable comments and construction recommendations, which were very helpful for revising and improving our paper. We have studied the comments carefully prior to the revision of our paper. All changes in the revised manuscript have been marked in the attached version of the text. We hope that the revised version of the manuscript will find the current version of our manuscript acceptable for publication.

The manuscript needs major revision.

Answer: The has been read carefully again and many corrections were made in each chapter, in line with the Reviewer's suggestion and recommendations. All changes suggested by the Reviewer are marked orange, while additional changes are marked in blue.

Extensive editing of English language and style required. Many sentences must be reformulated.

Answer: Within the limits of skills, linguistic and stylistic errors at work have been corrected.

Many experimental details must be added (characterization methods; thermograms).

Answer: The necessary correction have been made additionally in the Supplementary Materials FTIR spectra with marked wavelengths, thermograms and SEM photos at various magnifications for all tested materials were added.

The legend of figures must be revised.

Answer: The legend in all figures has been enlarged, darkened and clearer.

FTIR  and SEM figures must be revised as suggested.

Answer: The legend in all figures has been enlarged, darkened and clearer.

The authors confused urea with urine.

Answer: The error was corrected throughout the manuscript.

Authors must give the correct meanings of all physical quantities in equation 3.

Answer: All units were assigned to the appropriate data, the missing variables were completed.

Authors must be consequent with the abbreviations (FTIR or FT-IR, etc.).

Answer: The error was corrected throughout the manuscript.

Authors mentioned that they obtained nanocomposites, but they commented SEM images of hydrogels obtained and concluded that they have dimensions ranging between 6 and 600 micrometers, so it is a contradiction. Authors must revise this aspect.

Answer: Me must admit that it was exaggeration to use the term “nanocomposites” in this case, so we have removed the term from the manuscript

Authors must pay attention to all yellow highlighted paragraphs/ words.

Answer: All the Reviewer's suggestions were taken into account and corrected in the prescribed manner.

All the sections must be revised.

Answer: The entire manuscript was carefully reread. Many changes have been introduced in each of the chapters, as suggested by the Reviewers.

I made some corrections/suggestions/comments in the attached manuscript.

Answer: We would like to thank for the effort of indication our mistakes. It was very helpful guidance to improve our manuscript and we do appreciate it. Every improvement / suggestion / comment / has been incorporated and applied throughout the manuscript.

Round 2

Reviewer 1 Report

In this revised version, the authors made great improvements according to the comments and suggestions of both referees. It is suitable for publication now.

Author Response

Dear Reviewer,

On behalf of myself and the co-author, thank you for the positive review.

Best regards
Elżbieta Czarnecka

Reviewer 2 Report

Dear Authors,

The manuscript has been visibly improved, but it may be published after a minor revision.

I made some minor corrections/suggestions/comments in the attached manuscript. Authors must pay attention to all yellow highlighted paragraphs/ words.

Some sentences must be reformulated or must be removed.

Regarding Figure 3 (FTIR spectra), on Oy) axis, authors must insert:
Transmittance (%), and on Ox) axis, the wavenumber range must be: 4000 to 400, not 400 to 4000!

In reference [1], page 18, the authors must insert a link.

The article is interesting and can be published after a minor revision.

Author Response

On behalf of my co-authors and myself I want to express our thanks to the Reviewers for the valuable comments and construction recommendations, which were very helpful for revising and improving our paper. We have studied the comments carefully prior to the revision of our paper. All changes in the revised manuscript have been marked in the attached version of the text. We hope that the revised version of the manuscript will find the current version of our manuscript acceptable for publication.

Comments from the editors and reviewer:

The manuscript has been visibly improved, but it may be published after a minor revision.

I made some minor corrections/suggestions/comments in the attached manuscript. Authors must pay attention to all yellow highlighted paragraphs/ words.

Answer: The whole manuscript has been carefully reread. All suggestions / comments / corrections marked in yellow have been corrected.

Some sentences must be reformulated or must be removed.

Answer: All the necessary changes of the sentences has been made along with the removal of unnecessary fragments.

Regarding Figure 3 (FTIR spectra), on Oy) axis, authors must insert:
Transmittance (%), and on Ox) axis, the wavenumber range must be: 4000 to 400, not 400 to 4000!

Answer: The name of the y axis has been corrected and the scale of FTIR spectra from 4000 to 400 cm-1 has been changed.

In reference [1], page 18, the authors must insert a link.

Answer: A link to the source of the information has been added.

The article is interesting and can be published after a minor revision.
